# Pharmacogenetics of Type 2 Diabetes—Progress and Prospects

**DOI:** 10.3390/ijms21186842

**Published:** 2020-09-18

**Authors:** Yulia A. Nasykhova, Ziravard N. Tonyan, Anastasiia A. Mikhailova, Maria M. Danilova, Andrey S. Glotov

**Affiliations:** 1Department of Genomic Medicine, D.O. Ott’s Institute of Obstetrics, Gynecology and Reproductology, 199034 Saint-Petersburg, Russia; yulnasa@gmail.com (Y.A.N.); ziravard@yandex.ru (Z.N.T.); anamikhajlova@gmail.com (A.A.M.); elenamariamassa@gmail.com (M.M.D.); 2Laboratory of Biobanking and Genomic Medicine, Saint-Petersburg State University, 199034 Saint-Petersburg, Russia

**Keywords:** type 2 diabetes, metformin, sulfonylureas, meglitinides, DPP-4 inhibitors, GLP1R agonists, SGLT-2 inhibitors, polymorphism, pharmacogenetics

## Abstract

Type 2 diabetes mellitus (T2D) is a chronic metabolic disease resulting from insulin resistance and progressively reduced insulin secretion, which leads to impaired glucose utilization, dyslipidemia and hyperinsulinemia and progressive pancreatic beta cell dysfunction. The incidence of type 2 diabetes mellitus is increasing worldwide and nowadays T2D already became a global epidemic. The well-known interindividual variability of T2D drug actions such as biguanides, sulfonylureas/meglitinides, DPP-4 inhibitors/GLP1R agonists and SGLT-2 inhibitors may be caused, among other things, by genetic factors. Pharmacogenetic findings may aid in identifying new drug targets and obtaining in-depth knowledge of the causes of disease and its physiological processes, thereby, providing an opportunity to elaborate an algorithm for tailor or precision treatment. The aim of this article is to summarize recent progress and discoveries for T2D pharmacogenetics and to discuss the factors which limit the furthering accumulation of genetic variability knowledge in patient response to therapy that will allow improvement the personalized treatment of T2D.

## 1. Introduction

Type 2 diabetes mellitus (T2D) is a common chronic condition characterized by insulin resistance and beta-cell dysfunction that can seriously impair the overall quality of life. T2D currently affects more than 400 million people throughout the world and it is projected 552 million cases of T2D by the year 2030 [1]. T2D may lead to blindness, kidney failure, heart attacks, stroke and lower-limb amputation and can result in lower life expectancy by 5–10 years. In clinical practice it is often observed that T2D patients who receive identical antidiabetic regimens have significant variability in glycated hemoglobin (HbA1c) level, glycemic control, drug efficacy and tolerability and incidence of side effects [2]. Interindividual variation may be caused by numerous factors, such as genetic factors, physical inactivity, hypertension, age, gender and others [3]. Pharmacogenetics is a study focused on the variation in human genome and its influence on individual drug response, drug efficacy and toxicity. The discovery of genetic determinants modulating glycemic response might provide a clue to the treatment mechanism of the T2D and finally could advance the development of personalized treatment.

Advances in technology of genotyping and sequencing, implementation of recent genetic knowledge gained from the Human Genome Project and national genomes projects, as well as the development of robust statistical methods have allowed genome-wide association study (GWAS) and NGS to become the excellent tools for discovering common and rare genetic variants associated with particular phenotype. The modern powerful technologies of genotyping and sequencing have increased dramatically the genetic knowledge that provides a fascinating opportunity to use this information to predict the occurrence of disease and to identify subgroups of patients for whom therapies will have the greatest efficacy or the least adverse effect. To date, several studies including GWAS have been performed to investigate the influence of genetic polymorphisms on the therapeutic response in individuals with T2D. However, they have demonstrated contradictory results.

In this review, we summarize the evidences for association of genetic variants with therapeutic response in T2D patients to the common drugs including metformin, DPP-4 inhibitors/GLP1R agonists, sulfonylureas/meglitinides and SGLT-2 inhibitors, as well as we discuss limitations and future directions of pharmacogenetics in the field of T2D.

## 2. Pharmacogenetic Studies in Type 2 Diabetes

Type 2 diabetes mellitus is a chronic metabolic disease resulting from insulin resistance and progressively reduced insulin secretion, which leads to impaired glucose utilization, dyslipidemia and hyperinsulinemia and progressive pancreatic beta cell dysfunction. T2D etiology is known to have a significant genetic component that is confirmed by family- and twin-based studies [4]. The development of high-throughput and affordable genotyping technologies, statistical tools and computational software has led to the remarkable progress in the search for genetic associations in recent years [5,6,7,8,9]. Since the first genome-wide association study (GWAS) for T2D identified novel susceptibility loci in 2007, more than 100 T2D susceptibility loci have been discovered [10].

Over the past decade there has been a rapid development of antihyperglycemic agents. One of the main strategies, aimed to improve the quality of therapy, might be an individualized approach which is facilitated by pharmacogenetic studies. Pharmacogenomics is a branch of pharmacology that studies the influence of genetic variation on drug response in patients by correlating gene expression or single-nucleotide polymorphisms with a drug efficacy or toxicity [11]. The genetic variability of therapy response was recently shown in several independent studies for the common drugs used for T2D treatment, including biguanides (metformin), DPP-4 inhibitors/GLP1R agonists and sulfonylureas/meglitinides. Establishing the mechanism of drug-gene interactions may give an attractive opportunity to translate it into clinical practice—this might help stratifying the patient groups, decision-making on the therapeutic approach, reducing the rates of side effects and thereby improving the quality of treatment of T2D.

## 3. Genes Associated with Metformin Response

Metformin is an oral hypoglycemic agent and the only member of the biguanide class of drugs used in clinical practice. Metformin is commonly used as the first-line medication for the T2D treatment according to the recommendations of clinical guidelines [12,13]. Despite the popularity of metformin in diabetes treatment, the exact mechanism underlying the glucose level-lowering effects of this medication still remains poorly understood [14]. However, some metformin effects are undeniable (e.g., suppression of enhanced basal endogenous glucose production in patients with T2D through a 25–40% decrease in the hepatic gluconeogenesis rate that is hypothesized to be caused via activation of AMPkinase (AMPK) and direct inhibitory effects on mitochondrial function) [15]. Metformin is able to improve insulin sensitivity and insulin-stimulated glucose uptake in skeletal muscle. Metformin is not metabolized in the body and is excreted unchanged in the urine through active tubular secretion in the kidney [16]. Glucose lowering effect of metformin was shown to have a wide interindividual variability [17]. Moreover, metformin therapy is associated with more gastrointestinal symptoms (range 2–63% in different clinical trials) than most other oral antidiabetic agents. This may cause the premature termination of therapy in approximately 4% of cases [18,19]. The polymorphisms of genes involved in the various metabolic pathways of metformin can be assumed to cause different reactions as well as to develop side effects in patients who use drug therapy. The better understanding of metformin action mechanisms might certainly allow creating more tailored and precise treatment of T2D. Genes associated with therapeutic responses to metformin were summarized in Table 1.

To investigate genetic patterns of the glycemic response to metformin development, three full-scale genome-wide studies have been performed in different cohorts. In 2011, the first GWA study was completed in a GoDART (Genetics of Diabetes Audit and Research Tayside) cohort of European ancestry, including 1024 Scottish patients affected by T2D. Fourteen SNPs with a p-value < 1 × 10^−6^ were mapped to a 340 kb strong LD block on chromosome locus 11q22. Allele *C* of rs11212617 was found to be associated with treatment success estimated by ability to achieve an HbA1c < 7% in the 18 months after starting metformin treatment [OR = 1.64, 95%CI 1.37–1.99]. Then a variant rs11212617 was genotyped in 2 independent cohorts—GoDARTS cohort of 1783 metformin-treated patients and 1113 UK patients from UKPDS (UK Prospective Diabetes) cohort. A significant association with response to metformin therapy was found in both replication studies. The combined p-value for association of therapy and rs11212617 was statistically significant (*p* = 2.9 × 10^−9^). Variant rs11212617 is located in a large block of linkage disequilibrium, including a set of genes among which the *ATM* gene seems to be the most perspective candidate gene [20]. The *ATM* gene is located in 11q22.3 and belongs to the phosphatidylinositol 3-kinase family of proteins that responds to DNA damage by phosphorylating key substrates involved in DNA repair and/or cell cycle control. Mutations in *ATM* gene are implicated to have a causative role in ataxia telangiectasia.

An association between rs11212617 and metformin response was confirmed on several independent cohorts such as Chinese Han population and population of Western Saudi Arabia [22,23]. In a meta-analysis, including users of metformin selected from the Diabetes Care System West-Friesland (DCS), the Rotterdam Study from the Netherlands and the CARDS Trial from the UK, SNP rs11212617 was shown to be associated with treatment success. However, in individual cohorts of this study (the Rotterdam Study and CARDS) no significant association was observed (*p* = 0.15 and *p* = 0.86, respectively). An additional meta-analysis, performed in combination with data from the previously described GoDARTS and UKPDS, achieved the level of statistical significance (combined OR of 1.25, 95% CI 1.13–1.38, *p* = 7.8 × 10^−6^) [21]. Subsequently, no significant difference between response to metformin in patients with T2D and rs11212617 was observed in a set of studies [44,45]. The inconsistency between the different studies can be assumed to be resulted both from the features of the cohort—its ethnic composition, samples size as well as approaches used to patients’ selection and examination. The association of rs11212617 with metformin treatment response was not replicated in patients with impaired glucose tolerance involved in the Diabetes Prevention Program (DPP). Florez and colleagues supposed that the effect of metformin treatment could be more pronounced in individuals with a higher level of HbA1C in the disease setting. Moreover, the statistical power in this research may have been reduced due to the multiethnicity of the studied sample [46].

In 2016, the Metformin Genetics (MetGen) Consortium performed a three-stage GWAS, consisting of 13,123 participants of different ancestries. According to the research, *C*-allele of rs8192675 variant in the intron of *SLC2A2* gene was associated with 0.17% (*p* = 6.6 × 10^−14^) greater metformin-induced HbA1c in 10,577 participants with European ancestry. In the meta-analysis of 13,123 participants of any ancestry (European, Latino, African American) no genetic heterogeneity between different ethnic groups was observed (Phet > 0.29), despite the frequency of the *C*-allele of rs8192675 varied widely [24]. *SLC2A2* encodes GLUT2, an integral plasma membrane glycoprotein of the liver, islet beta cells, intestine and kidney epithelium. GLUT2 mediates facilitated bidirectional glucose transport. The association of rs8192675 with metformin response has been replicated in the German Diabetes Study [25].

To test the association of genetic variants with change of HbA1c level in metformin response treatment a GWAS approach was used in a large cohort of individuals with T2D in the cohort of Action to Control Cardiovascular Risk in Diabetes (ACCORD). The founded significant variants were analyzed in four replication cohorts. Variants rs254271 in *PRPF31* and rs2162145 in *CPA6* were associated with worse and better metformin response, respectively (*p* = 3.79 × 10^−6^, beta = 0.16; *p* = 4.04 × 10^−6^, beta = −0.197) and meta-analysis in independent cohorts displayed similar associations with metformin response (*p* = 1.2 × 10^−8^ and *p* = 0.005) [26]. Previous GWA studies have identified rs11212617 in the *ATM* gene and rs8192675 in *SLC2A2* gene as being associated with metformin response. Interestingly, these findings were not replicated in this study [20,24].

The earliest genetic studies of association between genetic variant and a particular phenotype were focused on family-based linkage analysis and analysis of candidate genes in small-size groups of patients. But this approach may be successful only for identification of genetic variants with large effects. Selection of candidate genes for study was most often based on knowledge of the gene’s biological functional impact on the trait or disease. Metformin is a hydrophilic molecule and drug transporters have the major role in its pharmacokinetics [47]. Given the importance of this interaction, genetic variants of drug transporter genes were the logical candidates to be investigated.

### 3.1. SLC22A1

*SLC22A1* gene encodes an organic cationic transporter 1 (OCT1) which is the most studied transporter regarding association of genetic variants with metformin response. OCT1 is a member of polyspecific organic cation transporters family expressed in the liver, kidney and intestine that are critical for the elimination of many endogenous amines as well as a wide array of drugs and environmental toxins [48]. OCT1 is responsible for hepatic uptake of metformin and *SLC22A1* polymorphisms can modulate metformin uptake and response in cells [29]. In *SLC22A1* gene knockout mice, the liver concentration of metformin was 30 times lower than in mice with normal functioning OCT1 transporters [30]. There were many studies of the *SLC22A1* genetic polymorphisms in different populations but results on the effects of *SLC22A1* polymorphisms on metformin therapeutic responses were rather contradictory [49]. Shu and colleagues suggested that genetic variation in *SLC22A1* may contribute to variation in response to metformin and they performed a series of experiments to analyze whether variants rs12208357 (*R61C*), rs34130495 (*G401S*), rs72552763 (*420del*) and rs34059508 (*G465R*) contribute to reduced therapeutic response to metformin clinically. In this study the healthy volunteers with at least one of four reduced function variants had significantly higher plasma glucose levels for most of the sampling time points during the 180-min OGTT than those carrying only reference *SLC22A1* alleles [29].

Shikata and colleagues showed association of *−43T > G* and rs628031 (*Met408Val*) of metformin response as negative and positive predictors, respectively [50]. In some cohorts, in *A/A* genotype carriers of rs628031 (*Met408Val*) were shown to have a significant reduction in HbA1c levels in comparison with those with the heterozygous genotype (*A/G*). The association of rs628031 and increased response to metformin therapy was observed in Chinese population [21]. Meanwhile in Iranian [44], European [30,51] and Japanese [50,52] populations, no significant difference was found between groups of patients with different response to metformin [53]. In cohort of T2D patients from Latvia the variant *Met408Val* was shown to predispose to the occurrence of symptoms of metformin intolerance. Authors of the study performed a research to evaluate the role of SNPs in metformin transporter *SLC22A1 (OCT1)*, *SLC22A2 (OCT2)*, *SLC47A1 (MATE1*) and SLC47A2 *(MATE2)* genes with respect to side effects of metformin therapy. The minor alleles of two variants of *SLC22A1* were associated independently with the presence of side effects (rs628031, rs36056065) [28].

In the study of Genetics of Diabetes Audit and Research in Tayside Scotland (GoDARTS) database, included 1531 patients with T2D who used metformin, the two most common variants rs12208357 (*R61C*) and rs72552763 (*420del*) were analyzed. The researchers found no association of *SLC22A1* variants with metformin response. *R61C* and *420del* variants did not affect the initial HbA1c reduction, the chance of achieving a treatment target, the average HbA1c on monotherapy up to 42 months or the hazard of monotherapy failure [54]. In the Rotterdam Study in a population of 102 T2D patients 12 polymorphisms in the *SLC22A1* gene were analyzed but for only rs622342 variant a statistically significant association with change in the HbA1c level was found [30].

### 3.2. SLC22A2

*SLC22A2* gene encodes the Organic Cation Transporter 2 (OCT2) which is mainly expressed on the basolateral membrane of renal tubule cells. OCT2 transports metformin into the proximal tubule cells and together with *MATE1* (*SLC47A1*) and *MATE2* (*SLC47A2*) mediates the secretion of metformin into urine [55,56]. In Korean study it was shown, that genetic variations rs201919874 (*T199I*), rs145450955 (*T201M*) and rs316019 (*A270S*) of *SLC22A2* gene were associated with decreased renal clearance and increased plasma concentrations of metformin [34]. The intergenic variants rs3119309, rs7757336 and rs2481030, located between *SLC22A2* and *SLC22A3* were significantly associated with metformin inefficiency in European cohort. Found in this study polymorphisms were noncoding but they could be in linkage disequilibrium with any causal SNPs within coding/regulatory regions of *SLC22A2, SLC22A3* or even *SLC22A1*, resulting in altered transport activity or expression level in target tissues [35]. However, in an unselected sample of 103 healthy male Caucasian volunteers no significant association between renal metformin clearance and some variants of *SLC22A2* (rs10755577, rs17588242, *Val502*, rs315996, rs316019, rs17589858, *Thr130*, rs2928035, rs316024, rs316025, rs316026, rs3127573, rs533452 and rs662301) was determined [57]. In a recent study Al-Eitan and colleagues found no statistical significance for 10 studied variants (rs10755577, rs17588242, rs17589858, rs2928035, rs3127573, rs316024, rs316025, rs316026, rs533452, rs662301) in *SLC22A2* to have any effect on glycemic control [58].

### 3.3. SLC47A1

The *SLC47A1* gene encodes the MATE1 (multidrug and toxin extrusion 1) transporter. *SLC47A1* is expressed in the luminal membranes of renal proximal tubules and bile canalicular membranes of hepatocytes and is responsible for the efflux of cationic compounds, including metformin, from the cells [59]. Previous findings indicated that MATE1 plays an important role in the pharmacodynamics of metformin and that its dysfunction may cause a marked elevation in the metformin concentration in the liver [60].

In the research of Becker and colleagues 11 SNPs in *SLC47A1* gene and the 10-kbp-downstream region were studied. An intronic variant rs2289669 (G > A) was significantly associated with a 0.30% larger decrease in HbA1c level after start of metformin therapy [61]. This association was replicated in Iranian [36], Chinese [62] and European [37] populations. While in large-scale genotyping study performed by DPP the association of variant rs2289669 in *SLC47A1* gene with weaker metformin response was determined [31]. However, in some populations influence of rs2289669 to metformin response was not confirmed [16,63]. Evidence was also obtained that variants rs2289669 in *SLC47A1* gene and rs594709 in *SLC22A1* gene might function together in metformin efficacy [32].

### 3.4. SLC47A2

*SLC47A2* gene encodes the MATE2 (multidrug and toxin extrusion 2) transporter, expressed in apical domain of proximal and distal renal tubular cells. *SLC47A1* and *SLC47A2* are located in tandem on chromosome 17p11.2. MATE2 facilitates the disposition of metformin from renal tubular cells into urine. A nonsynonymous variant rs562968062 (*Gly211Val*) *SLC47A2* was associated with a complete loss of transport activity, mainly because of a decrease in MATE2 protein expression [39]. Choi and colleagues characterized variants of *SLC47A2* to determine their association with metformin response. In this study the two nonsynonymous *SLC47A2* variants (*485C > T* and *1177G > A*) were associated with significantly lower metformin uptake and reduction in protein expression level. A variant of the basal promoter region rs12943590 (*−130G > A*) in this study was associated with significant increase in promoter activity. The study showed that the homozygous individuals for rs12943590 (*−130G > A*) of *SLC47A2* had a markedly weaker response to metformin [40].

### 3.5. Other Genes Associated with Metformin Response

Jablonski and colleagues performed a study of effect of 1590 SNPs on incident diabetes and their interaction with response to metformin or lifestyle interventions in DPP cohort included 2994 participants. Besides the association of the previously discussed genes of drug transporters *SLC47A1* (rs2289669), *SLC22A1* (rs683369), *SLC22A2* (rs662301) statistically significant interactions between metformin response and genetic variants were also reported for genes encoding additional proteins associated with AMP-activated protein kinase–dependent inhibition of gluconeogenesis (*PRKAA1* (rs249429), *PRKAA2* (rs9803799), *PRKAB2* (rs6690158), *STK11* (rs741765), *PPARA* (rs4253652), *PPARGC1A* (rs10213440), *PCK1* (rs4810083)); insulin secretion (*ABCC8* (rs4148609), *KCNJ11* (rs7124355), *HNF4A* (rs11086926), *HNF1B* (rs11868513)); insulin sensitivity (*ADIPOR2* (rs758027), *CAPN10* (rs758027), *GCK* (rs2908289)); energy metabolism (*MEF2A* (rs4424892), *MEF2D* (rs6666307)) and other genes (*ITLN2* (rs6701920), *GCG* (rs6733736), *PKLR* (rs17367421), *PPARGC1B* (rs741579)) [31,64]. To investigate the possible association of the genetic variants in six genes (*PRKAA1*, *STK11*, *PCK1*, *PPARGC1A*, *HNF1A and CAPN10*), encoding the key molecules mediating the metformin pharmacodynamic effect, with the response to metformin treatment Tkáč and colleagues performed a study on a small cohort of patients with T2D. A reduction in HbA1c level after the 6-month treatment with metformin in carriers of minor *G*-allele rs3792269 in the *CAPN10* gene was determined. *CAPN10* gene encodes the calpain 10 [37]. Calpains represent a ubiquitous, well-conserved family of calcium-dependent cysteine proteases. Calpain 10 has been suggested to influence both insulin resistance and insulin secretion [65,66]. A variant rs784888 in the *SP1* gene has been reported to influence on metformin response [41]. *SP1* gene encodes a zinc finger transcription factor which may modulate the gene expression of MATE1 and OCT3, two important metformin transporters [39]. A separate transcription factor binding analysis using a transcription factor binding tool, performed by Goswami and colleagues, demonstrated that in addition to regulating the expression of *MATE1* and *OCT3*, SP1 may also modulate levels of OCT2 and MATE2, transporters that are known to play a very important role in metformin elimination from the kidney [41]. *FMO5* gene encodes a flavin-containing monooxygenase 5, able to catalyze oxygenation of nitrogen-containing drugs. *FMO5* is expressed in the kidneys and liver [67]. The *FMO5* gene is localized near *PRKAB2*, a known regulator of metformin response. A variant rs7541245 in the gene *FMO5* was found to be associated with altered metformin response [42]. Chen and colleagues investigated the glucose lowering effect of metformin in mice. After an intravenous dose of metformin, a 2-fold decrease in the apparent volume of distribution and clearance was observed in *OCT3* knockout compared with wild-type mice indicating an important role of OCT3 in tissue distribution and elimination of the drug. A variant rs2076828 in *SLC22A3 (OCT3)* gene was found to be associated with reduced response to metformin [43].

## 4. Genes Associated with Sulfonylurea/Meglitinides Response

Sulfonylureas are the oral agents widely used in T2D treatment, which mechanism of action is stimulation of insulin secretion as well as of basal insulin via binding to sulfonylurea receptor 1 (SUR1). SUR1 is a part of the ATP-sensitive K^+^ (KATP) channel. The interaction between sulfonylurea and SUR1 causes inhibition of the KATP channel, decreasing the K^+^ efflux and depolarization of the β-cells. This leads to opening of voltage dependent Ca^2+^ channels, eliciting Ca^2+^ influx and increasing in intracellular Ca^2+^. The spike of intracellular Ca^2+^ levels triggers insulin zymogen fusion with the plasma membrane and insulin secretion [2,68]. Meglitinides are short-acting glucose-lowering drugs for therapy of patients with type 2 diabetes. Meglitinides are insulin secretagogues that stimulate insulin release from the pancreatic β-cells [69]. At present, sulfonylureas and meglitinides are recommended to be used as additional antihyperglycemic drugs in the absence of adequate glycemic control from lifestyle modification and metformin monotherapy. In addition, sulfonylureas and meglitinides can be used as a first line antihyperglycemic treatment due to intolerance to metformin or its adverse effects. Polymorphisms of genes *ABCC8, CYP2C9, KCNJ11, NOS1AP, TCF7L2,* C*YP2C8, KCNQ1* and *IRS1* have been reported for altered therapeutic response to sulfonylureas/meglitinides. Genes associated with therapeutic responses to sulfonylureas/meglitinides were summarized in Table 2.

### 4.1. ABCC8

The gene *ABCC8* located at 11p15.1 encodes SUR1 which modulates the activity of KATP channel [102]. Variants in *ABCC8* and *KCNJ11* (*Kir6.2*) genes may decrease or abolish the metabolic sensitivity KATP channel function of β-cells that leads to an altered depolarization of the cell membrane and a persistent insulin secretion, resulting in therapeutic failure of sulfonylureas. Several SNPs within the *ABCC8* gene have been reported to be associated with the response to sulfonylurea treatment. Elbein and colleagues demonstrated the association of an intronic variant rs1799854 with reduced insulin secretion after tolbutamide infusion in nondiabetic relatives of T2D patients [70]. In a study of 115 patients with T2D in China a significant decrease in HbA1c in *G* allele carriers of rs757110 variant of *ABCC8* compared with *T/T* homozygous individuals has been shown [75]. The influence of rs757110 (*Ser1369Ala*) variant in *ABCC8* gene to antidiabetic efficacy of gliclazides was demonstrated in two independent cohorts of Chinese patients after 8-week of gliclazide therapy [76]. A strong linkage disequilibrium between *E23K* in *KCNJ11* and rs757110 (*Ser1369Ala*) in *ABCC8*, two common KATP channel variants, was described by Florez and colleagues [77]. In a study by Nicolac and colleagues, variants rs1799854 and rs1799859 in the *ABCC8* gene have been shown to be associated with sulfonylurea treatment efficacy in Caucasians [78].

### 4.2. KCNJ11

The subunit Kir6.2 of the KATP channel is encoded by *KCNJ11* (potassium voltage-gated channel subfamily J member 11) gene. Four poreforming subunits of the inwardly rectifying potassium channel Kir6.2 and four regulatory subunits of SUR1 form the pancreatic β-cell KATP channel [103]. Performed studies of association between variant *Lys23Glu* (rs5219) of *KCNJ11* gene and effectiveness of sulfonylurea therapy demonstrated controversial results. In one Caucasian population there was no association [79], while in other Caucasian populations a higher rate of secondary sulfonylurea failure in KCNJ11 *23Glu* allele carriers was determined [81,82,96]. The association of *KCNJ11* and *ABCC8* polymorphisms with better response to sulfonylurea therapy was shown in two studies on Asian populations [75,76]. In the Slovakian study, the carriers of the *KCNJ11 23Glu* allele had better therapeutic response to gliclazide [81]. Another variant rs5210, located in 3′-UTR region of *KCNJ11* gene, was associated with sulfonylurea response in Chinese population. In a study performed in two independent cohorts of T2D patients who used gliclazide the association of rs5210 *KCNJ11* with gliclazide response was shown [76].

### 4.3. CYP2C9

*CYP2C9* encodes a cytochrome P450 2C9 enzyme involved in the metabolic elimination of compounds including steroid hormones, fatty acids and some drugs. Rs1799853 and rs1057910 have been recently reported to determine lower CYP2C9 catalytic activity, leading to reduced SUF clearance [2,87]. In several pharmacokinetic studies two common variants (rs1799853 and rs1057910) of *CYP2C9* have been shown to be associated with therapeutic response to sulfonylureas [83,84]. In a large cohort Zhou and colleagues demonstrated that the loss-of-function alleles of rs1799853 and rs1057910 variants of *CYP2C9* were robustly associated with greater response to sulfonylureas in patients with type 2 diabetes. They determined a significantly higher reduction in HbA1c concentration in patients with two polymorphic *CYP2C9* alleles than in carriers of two wild-type alleles [86]. However, in other study the HbA1c level was not found to be influenced by *CYP2C9* genotype [104].

### 4.4. NOS1AP

The *NOS1AP* gene encodes the nitric oxide synthase I adaptor protein belonging to the responsible for nitric oxide generation class of enzymes. NOS1AP is a regulator of neuronal NOS (nNOS encoded by NOS1), one of the isoforms of NOS. It is believed that the nNOS enzyme may regulate intracellular calcium levels and inhibit the influx of Ca^2+^ through voltage dependent calcium channels. Thereby, nNOS might influence insulin release [88,105,106].

In the cohort of 619 participants, for carriers of *TG* or *GG* genotype of rs10494366 in the *NOS1AP* gene, glibenclamide was less effective in reducing glucose levels compared with patients with the *TT* genotype. Moreover, in this study it was shown that in users of glibenclamide the *TG* and *GG* genotypes were associated with an increased risk of mortality, In tolbutamide and glimepiride users the *TG* or *GG* genotypes were associated with a reduced risk of mortality [88]. In addition, the carriers of *TT* genotype in Chinese population had an increased effect of repaglinide on insulin resistance measured by HOMA index [89]. In Korean study no significance was found between rs10494366 in the *NOS1AP* gene and response on glimepiride treatment [107]. The effect of rs12742393 in the *NOS1AP* gene on repaglinide response was determined in Chinese patients with T2D. T2D patients with genotypes *AA* and *AC* of rs12742393 *NOS1AP* had a significant reduction in fasting plasma glucose levels compared with those with genotype *CC* [90].

### 4.5. TCF7L2

*TCF7L2* encodes a transcription factor involved in WNT signaling [108]. A predominant direct or indirect role of TCF7L2 on β-cell function has been suggested. Pearson and colleagues determined the association of two genetic variants *TCF7L2* (rs1225372 and rs7903146) with initial treatment success of sulfonylurea therapy in T2D patients. It was shown that with respect to rs1225372 12% of the diabetic population with *TT* genotype were twice as unlikely to achieve a target HbA1c < 7% within 1 year of treatment initiation compared to 42% of the population with *GG* genotype [93]. These findings were replicated in some other studies in Indian and European populations [87,91,92]. In Asian population genetic variant rs290487 of *TCF7L2* gene was found to influence the therapeutic efficacy. T2D patients with at least one *T* allele of *TCF7L2* rs290487 had an increased efficacy of repaglinide [94].

### 4.6. Other Genes Associated with Sulfonylurea/Meglitinides Response

There is some evidence of association of other genes with sulfonylurea response. C*YP2C8* encoding cytochrome P450 (CYP) 2C8 is responsible for the oxidative metabolism of many clinically available drugs from a diverse number of drug classes (e.g., thiazolidinediones, meglitinides, NSAIDs, antimalarials and chemotherapeutic taxanes). The polymorphism of C*YP2C8* (CYP2C8*2 and CYP2C8*3) was shown to be associated with increased metabolism of repaglinide [98,99]. Variant rs1801278 (*Gly972Arg*) in the insulin receptor substrate-1 (*IRS1*) gene has been shown to play a role in insulin signal transduction and therapeutic failure to sulfonylurea drugs [95,96,97]. *KCNQ1* polymorphisms are associated with repaglinide efficacy 101]. The association between variants in the *KCNQ1* gene and the effects of repaglinide and rosiglitazone was found in three independent studies [92,100,101].

## 5. Genes Associated with DPP4-inhibitors and GLP1 Receptor Agonists Response

According to the ADA (American Diabetes Association) and the EASD (European Association for the Study of Diabetes) standards of medical care in diabetes (2020), DPP4-inhibitors (DPP4I) and GLP1 receptor agonists (GLP1RA) are recommended as a second-line glucose-lowering agents in cases when patients require combination therapy for achieving adequate glycemic control or when metformin or sulfonylurea therapy is accompanied by episodes of hypoglycemia or another adverse reaction [13,109]. A few pharmacogenetic studies of second-line antidiabetic drugs have been conducted over the past decade. Most of them have been focused mainly on candidate genes involved in DPP4I and GLP1A pharmacokinetic mechanisms. Genes associated with therapeutic responses to GLP1A and DPP4I were summarized in Table 3 and Table 4.

GLP1 (glucagon-like peptide-1) and GIP (gastric inhibitory polypeptide) are hormones releasing after food intake and stimulating insulin secretion [120]. The mechanism of DPP4I action is to increase endogenous incretin hormones levels by inhibiting dipeptidyl peptidase IV, an enzyme responsible for rapid degradation of GLP1 and GIP [121]. Due to NH2-terminal-enzyme destruction and renal clearance, glucose-lowering effect of circulating incretin hormones is limited. Therefore, GLP1RA therapy overcomes the main limitation of native GLP1 and GIP [122].

### 5.1. GLP1R

Several candidate gene studies have been performed to assess the role of *GLP1R* gene variants in DPP4I and GLP1RA treatment efficacy. The potential role of genetic variability of *GLP1R* in the therapeutic response to GLP1RA was first demonstrated in 2010 by Sathananthan and colleagues. An increase of β-cell responsivity to GLP1 infusion in healthy homozygotes for the major allele of rs6923761 was demonstrated. Rs6923761 is a non-synonymous SNP which results in the substitution of serine for glycine at position 168 [113]. The study conducted by De Luis and colleagues has shown that the decrease of waist circumference, waist-to-hip ratio and systolic blood pressure were higher in polymorphic rs6923761 allele carriers compared to two wild type alleles carriers during liraglutide treatment. However, the decrease in basal glucose and HbA1c did not differ in carriers of both genotypes [111]. Similar results have been demonstrated in obese women with polycystic ovary syndrome carrying genotypes *GA* or *AA* [110]. It should be noted though, that according to Chedid and colleagues, there was no statistically significant difference in weight loss between wild type and polymorphic allele carriers after exenatide and liraglutide treatment. Nevertheless, researchers have found prolonged gastric emptying half-life in *GA*/*AA* genotypes [112].

As has been previously reported by Javorsky and colleagues, T2D patients with *Ser/Ser* genotype had a lower reduction in HbA1c during DPP4I treatment as well [124]. Mashayekhi and colleagues also reported that subjects with one or two copies of the wild-type allele had increased postprandial glucose excursion as compared with *Ser/Ser* subjects while undergoing sitagliptin treatment [125]. Results of the study have been successfully replicated in a recent research which has proved an association between smaller glycemic responses to 6 month gliptin therapy in diabetic patients with a missense variant rs6923761 in the *GLP1R* gene [126].

Another missense SNP rs3765467 results in substitution of glutamine for arginine in position 131. The data obtained by researchers regarding this polymorphism were very contradictory. One of the first reports on this SNP showed increased β-cell responsivity to GLP1 infusion in healthy heterozygotes for the minor allele when compared with homozygotes for the major allele [113]. On the contrary, in the study of Yu and colleagues mutant *T* allele of rs3765467 was marginally associated with a less reduction in HbA1c level [115]. Finally, another study did not find any significant difference in clinical response to exenatide except for a lower standard deviation of plasma glucose level change [114]. As for DPP4I, it has been shown, that minor allele carriers had higher reduction in HbA1c during treatment. 

Rs10305420 polymorphism has also been described in two articles. The minor allele has been shown to be associated with a less reduction in HbA1c level and weight while undergoing exenatide and liraglutide treatment [110,115].

### 5.2. TCF7L2

Genetic variations in *TCF7L2* gene have been shown to influence the efficacy of DPP4I and GLP1RA therapy as well. *TCF7L2* gene encodes a transcription factor required for maintaining both basal and GLP1RA-induced proliferation of beta cells [137]. Expression of the GLP1R and GIP-R also depends on transcriptional activity of *TCF7L2* [138]. The SNP rs7903146 is one of the most studied polymorphisms associated with type 2 diabetes within *TCF7L2* gene [139]. It has been demonstrated that *T* allele carriers had lower decreases in HbA1c and 2-h plasma glucose levels when treated with DPP4I [127]. However, according to a recent study by Ferreira and colleagues, *T* allele carriers are more responsive to GLP1RA treatment [116].

### 5.3. DPP4

The enzyme encoded by *DPP4* gene is a target molecule for DPP4I, therefore polymorphisms of the gene can affect the drug efficacy. The first study conducted to evaluate the effect of DPP4 gene on DPP4I therapeutic response did not reveal any significant association [140]. Afterwards, Wilson and colleagues demonstrated that carriers of *T* allele of rs2909451 and *C* allele of rs759717 have an increased DPP-4 activity during sitagliptin treatment [128].

### 5.4. KCNQ1

*KCNQ1* is a gene encoding an alpha-subunit of a voltage-gated potassium channel [141]. This channel is known to be responsible for regulation of insulin secretion [142]. The data obtained by Liu et al. demonstrated that KCNQ1 channels inhibition resulted in increased insulin secretion and increased GLP1 level in mice [143]. Besides this, SNP rs151290 was associated with GLP-1 increase after oral glucose tolerance test in non-diabetic subjects [144]. Unfortunately, this association has not been confirmed by subsequent research [145]. Nevertheless, *KCNQ1* gene certainly requires further evaluation of its effect on pharmacogenetics of incretin mimetics. Rs163184 located in intron of *KCNQ1* gene has been proven to be a type 2 diabetes susceptibility factor by GWAS. Also it was connected with therapeutic response to sulfonylurea treatment [146,147]. The association of the G-allele with smaller reduction in HbA1c level while DPP4I treatment has been demonstrated twice in independent studies [126,129].

### 5.5. Other Genes Associated with DPP4-Inhibitors and GLP1 Receptor Agonists Response

While the most studied associations were considered above, there is evidence of other genes related to response to GLP1RA and DPP4I therapy. These genes require further study and novel researches in order to confirm the found associations and to fully understand the underlying mechanisms of their involvement in DPP4I and GLP1RA pharmacogenetics.

For instance, pro-inflammatory cytokine encoded by *IL6* gene is known to stimulate GLP1 secretion [148,149]. The correlation between *IL6* polymorphisms and type 2 diabetes susceptibility has been described by many researchers but remains controversial. Thus, Qi and colleagues could not find any substantial association of type 2 diabetes risk with both circulating interleukin level and polymorphisms in *IL6*, including rs1800795 [150]. On the contrary, the *GC* and *CC* genotypes of the *IL6* rs1800795 have been shown to be protective factors against type 2 diabetes [151]. Matsui et al. performed a study on the effects of SNPs in the *IL6* promoter region on response to DPP4I. The results showed that diplotype rs1800796 allele *G* and rs2097677 allele *A* might contribute to DPP4I responsiveness in case the carrier is physically active [130].

According to Hugill et al., mutations in *KCNJ11* result in an impaired incretin response in mice after intraperitoneal glucose tolerance testing [152]. A strong association between poor DPP4I efficacy and allele *T* of the *KCNJ11* rs2285676 has been demonstrated afterwards [131].

Type 2 diabetes susceptibility variant rs7202877 located in an intergenic region between *CTRB1* and *CTRB2* has also been found to be connected with a smaller reduction in HbA1c in allele *G* carriers while DPP4I treatment but not GLP1RA [132,153].

Rs738409 in *PNPLA3* gene has been confirmed to have effect on diabetes, insulin, insulin resistance and body mass index too [154]. A positive correlation between reduction in HbA1c and liver transaminases levels and also a better improvement in total cholesterol, triglycerides and hyaluronic acid has been observed in allele *G* carriers treated with alogliptin [133].

Polymorphism located within the fourth intron of *PRKD1* gene has found to be closely related to DPP4I treatment efficacy [134], what can be explained by serine/threonine-protein kinase D1 participation in increasing of insulin secretion in mice and human islets [155].

The potential effect of P-glycoprotein 1, also known as multidrug resistance protein 1 and encoded by *ABCB1* gene, on DPP4I pharmacology was first investigated by Aquilante et al. Rs2032582, rs1045642 and rs1128503 in the gene were analyzed in healthy volunteers. These SNPs have been shown to have no impact on sitagliptin pharmacokinetics [156]. However, according to more recent research, allele *G* carriers of the last polymorphism were more responsive to DPP4I therapy [135].

As known, N-acetyltransferase 2, encoded by *NAT2* gene, plays important role in activation and deactivation of arylamine and hydrazine drugs and carcinogens and thereby influences the toxicity and efficacy of some medications [157]. The deficiency of *NAT2* gene may result in decreased insulin sensitivity [158]. The association of SNP rs1041983 polymorphism with DPP4I efficacy has been analyzed by Iskakova and colleagues. Carriers of the allele *T* had a higher probability of not reaching the target HbA1c level [135].

An association of *CDKAL1* gene rs7754840 and rs7756992 with type 2 diabetes susceptibility was initially discovered from a GWAS [159,160] and confirmed by subsequent candidate gene studies [161,162]. Both of these polymorphisms have been shown to affect the HbA1c reduction level while DPP4I therapy [136]. Another SNP in *CDKAL1* gene, rs10946398, is associated with impaired insulin secretion; therefore it would be valuable to further investigation of this polymorphism in terms of impact on DPP4I and GLP1RA pharmacogenetics [163].

Cannabinoid receptor type 1 encoded by *CNR1* gene is expressed in many human tissues. Genetic variants at *CNR1* are known to be associated with increased HOMA-IR [164]. According to Moss et al., CNR1 agonists can modulate incretins regulation by inhibiting GIP secretion [165]. The association of allele *G* of rs1049353 with decreased total cholesterol and LDL-cholesterol has also been demonstrated in obese diabetic patients during GLP1RA treatment [117].

Another gene with a proven association with type 2 diabetes is *SORCS1* [166]. The allele *G* of rs1416406 has been shown to be connected with higher reduction in proinsulin/insulin ratio in patients treated with exenatide [118].

Mutations in *WFS1*, the wolframin gene, are responsible for the development of Wolfram syndrome, an autosomal recessive disease characterized by a variety of disorders including diabetes mellitus [167]. In a mice study researchers have demonstrated that treatment with GLP1RA may improve beta cell function, show a neuroprotective effect and even preserve from glucose intolerance in case of preventive therapy [168,169,170,171]. As GLP1RA may be considered as a potential therapy for wolfram syndrome treatment, analysis of association between *WFS1* gene polymorphism and GLP1RA efficacy becomes reasonable. Thus, Pereira and colleagues demonstrated higher body weight loss in allele *A* carriers of rs10010131 [119].

## 6. Genes Associated with SGLT-2 Inhibitors Response

Sodium-glucose cotransporter-2 (SGLT2) inhibitors are the class of anti-hyperglycemic agents that has been approved to use for the treatment of type 2 diabetes relatively recently. SGLT2 inhibitors reduce the renal tubular glucose reabsorption that subsequently results in reduction of plasma glucose concentrations in blood. Moreover, SGLT2 inhibitors may have additional favorable effects including the reducing of blood pressure and weight [172]. To date the information about the role of genetic polymorphisms in response to SGLT-2 inhibitors is quite limited and additional studies are required to reveal the mechanisms of the potential effects.

Franke and colleagues performed a study to identify in vitro human enzymes and tissues for major glucuronidated metabolites of canagliflozin and subsequently to access the impact of genetic variations on its pharmacokinetics. They demonstrated the role of *UGT1A9*3* allele (rs72551330) in canagliflozin pharmacokinetics [173]. *UGT1A9* gene encodes a UDP-glucuronosyltransferase, an enzyme of the glucuronidation pathway that transforms lipophilic molecules such as hormones, steroids and drugs into soluble and excretable metabolites. Some SGLT-2 inhibitors (canagliflozin) can be glucuronidated by UGT enzymes, thereby polymorphisms in the genes encoding these drug-metabolizing enzymes could potentially influence its response. *SLC5A2* gene encodes a member of the sodium glucose cotransporter family. The encoded protein is the major cotransporter involved in glucose reuptake in the kidney. Some inactivating mutations in *SGLT2* were shown to be associated with familial renal glucosuria characterized by considerable reduced tubular reabsorption of glucose resulting in elevation of renal glucose excretion [174]. It was shown that *SGLT2* knockout mice have improved plasma insulin levels [175]. Therefore, *SGLT2* is seemed to be a promising target for pharmacogenetic studies. In a large study of 2229 subjects at increased risk for type 2 diabetes the impact of common *SLC5A2* variants (rs9934336, rs9924771, rs3813008 and rs3116150) on empagliflozin response was analyzed. However, in this cross-sectional study, no significant association between the tested SNPs and plasma glucose concentrations, insulin sensitivity/resistance, insulin release, body fat or systolic blood pressure was detected. Only one SNP, rs3116150, showed a nominal association with plasma glucose and blood pressure [176].

## 7. Discussion

Pharmacogenetics provides a fascinating opportunity to improve patient care by optimizing the choice and dosage of medications, reducing the risk of adverse effects and, thereby, implementing the principles of personalized medicine. However, results of the recent pharmacogenetic studies of treatment effect in patients with T2D demonstrated the existence of certain limitations that might impede the implementation of new knowledge in clinical practice. The crucial limitation is that the most findings despite achieving the considerable level of statistical significance in the original study have failed to be replicated in other cohorts or even they show controversial results. To a large extent this inconsistency can be explained by the differences in study design including the size and characteristics of cohort, type of study, methods of analysis and methods of estimation of the treatment effectiveness.

In that context, the choice of study design may play an important role in the quality, execution and interpretation of results of biomedical research. A prospective study is preferable to a retrospective study. Retrospective studies are more prone to different biases, particularly recall bias [177]. In addition, it should be noted that in most of the analyzed studies of candidate genes the sample consisted of a hundred or even dozens of individuals and considering the contribution of a genetic variant may be quite small, it can be anticipated that the insufficient sample size may be the reason of getting false results. According to the formula for the estimation of the sample size [178] for study of an SNP the cohort should include at least 287 individuals to get the statistically adequate results [179,180]. However, the presence of possible confounding factors must be taken into account and thereby, a larger sample is required to achieve significance. Apart from the sample size other characteristics of the sample are of great importance such as age, duration of diabetes, age at diagnosis, glycemic control, gender, comorbidities, used drugs, baseline HbA1c. Differences in cohort characteristics may lead to incorrect interpretation of the obtained results and cause hasty extrapolation of conclusions for all patients. Moreover, analysis of a number of studies conducted in large cohorts demonstrated that their results did not always replicate. In GWA study performed by Rotroff and colleagues the variants rs11212617 in the *ATM* and rs8192675 in *SLC2A2* previously identified as being significantly associated with metformin response have shown no association with metformin treatment (P > 0.1) [20,22,26]. Rotroff and colleagues suggested that this failure to replicate the findings could be caused by cohort differences and, in particular, the sample size of the ACCORD cohort and the use of concomitant medications [26]. It is necessary to take into account genetic heterogeneity in different ethnic populations and causal pathways, as well as the fact that relative contribution of polymorphisms to these pathways may differ between populations. Perhaps, this leads to the differences in results obtained in studies on different populations. The inconsistency in results obtained on GoDARTS cohort of European ancestry and in ACCORD study can be assumed to be caused, furthermore, by population differences.

In research of pathogenesis of any condition the possible gene-gene interaction should be taken into account as well. Results of a study of Christensen and colleagues found a counteracting effect of the genetic variations *SLC22A1* rs316019 and *SLC47A1* rs2252281 on the renal elimination of metformin [181]. Xiao and colleagues demonstrated an interaction between *SLC47A1* rs2289669 and *SLC22A1* rs594709, which affects the blood glucose, insulin level, insulin resistance and blood lipid improvement after metformin treatment in Chinese patients [32].

It should be mentioning the currently actively investigated microRNAs, a class of small non-protein-coding regulatory molecules that could serve as potential biomarkers for the diagnosis of type 2 diabetes. Recently, the impact of microRNAs on diabetes pathogenesis was shown in several studies that makes them an interesting target for pharmacogenetic studies. The effect of microRNAs on statins, anticoagulants, cytotoxic drugs pharmacogenetics is well studied by now [182]. Meanwhile, the effect of microRNAs on drug response during oral glycemic therapy is yet remains to be fully understand and further investigations are required in this field. As microRNA regulates gene expression by binding to mRNA, polymorphisms in the microRNA binding sites, as well as polymorphisms within the microRNA encoding genes, can lead to changes in the expression of genes responsible for pharmacokinetics and pharmacodynamics of the drugs [183]. The possibility of using the microRNA level as a marker of efficacy of therapy is also being evaluated. For instance, Demirsoy and colleagues demonstrated altered expression profiles of 13 microRNAs after metformin treatment [184]. Moreover, microRNA levels can be modulated by GLP1 and GLP1RA [185]. Thus, microRNA-based strategy may help to improve approaches to assessing the quality of type 2 diabetes therapy.

Besides, DNA methylation and another epigenetic modifications of genes responsible for oral glycemic therapy response may also influence the diabetes pharmacogenetics. Aberrant DNA methylation of *OCT1* and genes involved in metformin pharmacokinetics might affect their hepatic expression, resulting in affected metformin transport [186].

Moreover, hyperglycemia in pregnancy is known to be associated with adverse long-term health outcomes both for mother and offspring with 2–5 fold increased risk of congenital anomaly, stillbirth and neonatal death compared with the general maternity population [187]. Studies of impact of genetic variability in drug efficacy and adverse effects in pregnant women with T2D also would be of great interest.

The findings indicates that distinct mechanisms underlined in the effects to drugs used for T2D treatment are likely multifactorial. Performed analysis of published articles suggest the plausible role of the genetic factors in the therapy response of common drugs for T2D treatment including polymorphisms of 31 genes in metformin response, polymorphisms of 8 genes in sulfonylureas/meglitinides response, polymorphisms of 12 genes in DPP-4 inhibitors response and polymorphisms of 5 genes in GLP1R agonists response. To confirm these findings, further studies on the large cohorts of well-examined patients in the different ethnic and homogeneous populations are required. In the future better understanding of pharmacogenetics of T2D including the knowledge of common and rare gene variants will allow development of personalized instruments for improved therapy.

## 8. Methods

A search has been performed using PubMed/Medline databases. The articles relevant to the research topic published until July 2020 were included in this review. The relevant information from the electronic database “Pharmgkb” [188] was also included. A search of the references cited in initially identified articles was performed.

## Figures and Tables

**Table 1 ijms-21-06842-t001:** Genetic variants that influence metformin therapy outcomes.

No.	Gene Symbol	Region	dbSNP ID	SNP	Alleles	Effect	References
1	*ATM*	11q22.3	**rs11212617**	Intronic	A/C	↑	[20,21,22,23]
2	*SLC2A2*	3q26.2	**rs8192675**	Intronic	A/G	↑	[24,25]
3	*PRPF31*	19q13.42–q13.42	rs254271	Intronic	C/A	↓	[26]
4	*CPA6*	8q13.2–q13.2	rs2162145	Intronic	T/A	↑	[26]
5	*SLC22A1*	6q25.3	**rs628031**	Missense *Met408Val*	A/G	↑ SE	[27,28]
			s12208357	Missense *Arg61Cys*	C/T	↓	[29]
			rs34130495	Missense *Gly401Ser*	A/G	↓	[29]
			rs622342	Intronic	C/T	↓	[30]
			rs683369	Missense *Leu160Phe*	G/C	↓	[31]
			rs36056065	Indel GTAAGTTG	-/GTAAGTTG	SE	[28]
			rs594709	Intronic	A/G	↑	[32]
			rs2282143	Missense *Pro341Leu*	C/T	↑	[33]
			rs72552763	IndelGAT	-/GAT	↓	[29]
6	*SLC22A2*	6q25.3	rs316019	Missense *Ala270Ser*	G/T	↓	[34]
			rs145450955	Missense *Thr201Met*	G/A	↓	[34]
			rs201919874	Missense *Thr199Ile*	C/T	↓	[34]
			rs3119309	Intergenic	C/T	↓	[35]
			rs7757336	Intergenic	G/T	↓	[35]
			rs2481030	Intergenic	A/G	↓	[35]
			rs662301	Non coding transcript	T/C	↓	[31]
7	*SLC47A1*	17p11.2	**rs2289669**	Intronic	G/A	↑↓	[31,32,36,37]
			rs2252281	5′ UTR	T/C	↑	[38]
8	*SLC47A2*	17p11.2	rs562968062	Missense *Gly211Val*	C/A	↓	[39]
			rs12943590	5′ UTR	G/A	↑	[40]
9	*PRKAA1*	5p13.1	rs249429	Intronic	C/T	↑	[31]
10	*PRKAA2*	1p32.2	rs9803799	Non coding transcript	G/T	↑	[31]
11	*PRKAB2*	1q21.1	rs6690158	Intronic	T/C	↓	[31]
12	*STK11*	19p13.3	rs741765	Intronic	T/C	↑	[31]
13	*PPARA*	22q13.31	rs4253652	Intronic	G/A	↓	[31]
14	*PPARGC1A*	4p15.2	rs10213440	Intronic	C/T	↓	[31]
15	*PCK1*	20q13.31	rs4810083	Intergenic	T/C	↑	[31]
16	*ABCC8*	11p15.1	rs4148609	Intronic	A/G	↑	[31]
17	*KCNJ11*	11p15.1	rs7124355	Intergenic	A/G	↓	[31]
18	*HNF4A*	20q13.12	rs11086926	3′ UTR	G/T	↓	[31]
19	*HNF1B*	17q12	rs11868513	Intronic	A/G	↑	[31]
20	*ADIPOR2*	12p13.33	rs758027	Intergenic	C/T	↓	[31]
21	*CAPN10*	2q37.3	rs3792269	Missense *Arg197Gly*	A/G	↑	[37]
22	*GCK*	7p13	rs2908289	Intronic	A/G	↓	[31]
23	*MEF2A*	15q26.3	rs4424892	Intergenic	G/A	↓	[31]
24	*MEF2D*	1q22	rs6666307	Intronic	T/A	↓	[31]
25	*ITLN2*	1q23.3	rs6701920	3′ UTR	A/G	↑	[31]
26	*GCG*	2q24.2	rs6733736	Intronic	G/A	↓	[31]
27	*PKLR*	1q22	rs17367421	Intronic	C/G	↓	[31]
28	*PPARGC1B*	5q32	rs741579	Intronic	G/A	↑	[31]
29	*SP1*	12q13.13	rs784888	Intronic	G/C	↑	[41]
30	*FMO5*	1q21.1	rs7541245	Intronic	C/A	↓	[42]
31	*SLC22A3*	6q25.3	rs2076828	Non coding transcript	C/G	↓	[43]

↑—increased response to therapy (in relation to the minor allele); ↓—reduced response to therapy (in relation to the minor allele); SE—side effect; bold font highlights the SNPs, associated with treatment response in 2 and more studies.

**Table 2 ijms-21-06842-t002:** Genetic variants that influence sulfonylureas/meglitinides therapy outcomes.

No.	Gene Symbol	Region	dbSNP ID	SNP	Alleles	Effect	References
1	*ABCC8*	11p15.1	**rs1799854**	Intronic	C/T	↓	[62,70,71,72,73,74]
			**rs757110**	Missense *Ala1369Ser*	T/G	↑	[75,76,77]
			**rs1799859**	Synonymous *Arg1273Arg*	G/A	↑	[72,78]
2	*KCNJ11*	11p15.1	**rs5219**	Missense *Lys23Glu*	C/T	↑ SF	[62,75,76,79,80,81,82]
			rs5210	3′ UTR	G/A	↑	[76]
3	*CYP2C9*	10q23.33	**rs1799853**	Missense *Arg144Cys*	C/T	↑	[83,84]
			rs9332239	Missense *Pro489Ser*	C/T	↓	[85]
			**rs1057910**	Missense *Ile359Leu*	A/C	↑	[86,87]
4	*NOS1AP*	1q23.3	**rs10494366**	Intronic	G/T	↑↓	[88,89,90,91]
			**rs12742393**	Intronic	A/C	↑	
5	*TCF7L2*	10q25.2	**rs7903146**	Intronic	C/T	↓	[69,87,91,92]
			**rs12255372**	Intronic	G/T	↑↓	[87,91,92,93]
			rs290487	Intronic	C/T	↑	[94]
6	*IRS1*	2q36.3	**rs1801278**	Missense *Gly972Arg*	G/A	↓SF	[95,96,97]
7	*CYP2C8*	10q23.33	**rs10509681 (*2)**	Missense *Lys399Arg*	T/C	↑	[98,99]
			**rs11572080 (*3)**	Missense *Arg139Lys*	G/A	↑	[98,99]
8	*KCNQ1*	11p15.4	rs2237892	Intronic	C/T	↑	[100]
			rs163184	Intronic	T/G	↓	[92]
			**rs2237895**	Intronic	A/C	↑	[100,101]

↑—increased response to therapy (in relation to the minor allele); ↓—reduced response to therapy (in relation to the minor allele); SE—side effect; SF—secondary failure; bold font highlights the SNPs, associated with treatment response in 2 and more studies.

**Table 3 ijms-21-06842-t003:** Genetic variants that influence GLP1RA therapy outcomes.

No.	Gene Symbol	Region	dbSNP ID	SNP	Alleles	Effect	References
1	*GLP1R*	6p21.2	**rs6923761**	Missense Gly168Ser	G/A	↑↓	[110,111,112,113]
			**rs3765467**	Missense Arg131Leu	C/T	↑↓	[113,114,115]
			**rs10305420**	Missense Pro7Leu	C/T	↓	[110,115]
			rs761386	Intronic	C/T	↑↓	[114]
2	*TCF7L2*	10q25.2	rs7903146	Intronic	C/T	↑	[116]
3	*CNR1*	6q15	rs1049353	Synonymous Thr453Thr	A/G	↑	[117]
4	*SORCS1*	10q25.1	rs1416406	Intronic	A/G	↑	[118]
5	*WFS1*	4p16.1	rs10010131	Intronic	A/G	↓	[119]

↑—increased response to therapy (in relation to the minor allele); ↓—reduced response to therapy (in relation to the minor allele); bold font highlights the SNPs, associated with treatment response in 2 and more studies.

**Table 4 ijms-21-06842-t004:** Genetic variants that influence DPP4I therapy outcomes.

No.	Gene Symbol	Region	dbSNP ID	SNP	Alleles	Effect	References
1	*GLP1R*	6p21.2	rs3765467	Missense *Arg131Gln*	C/T	↑	[123]
			**rs6923761**	Missense *Gly168Ser*	G/A	↑↓	[124,125,126]
2	*TCF7L2*	10q25.2	rs7903146	Upstream gene	C/T	↓	[127]
3	*DPP4*	2q24.2	rs2909451	Intronic	C/T	↓	[128]
			rs759717	Intronic	G/C	↓	
4	*KCNQ1*	11p15.4	**rs163184**	Intronic	T/G	↓	[126,129]
5	*IL6*	7p15.3	rs1800796	Upstream gene	G/*	↑ ^†^	[130]
			rs2097677	Intronic	A/*		
6	*KCNJ11*	11p15.1	rs2285676	3′UTR	A/T	↓	[131]
7	*CTRB1/2*	16q23.1	rs7202877	Intergenic	T/G	↓	[132]
8	*PNPLA3*	22q13.31	rs738409	Missense *Ile148Met*	C/G	↑	[133]
9	*PRKD1*	14q12	rs57803087	Intronic	A/G	↑↓	[134]
10	*ABCB1*	7q21.12	rs1128503	Synonymous *Gly412Gly*	A/G	↓	[135]
11	*NAT2*	8p22	rs1041983	Synonymous *Tyr94Tyr*	C/T	↓	
12	*CDKAL1*	6p22.3	rs7754840	Intronic	C/G	↓	[136]
			rs7756992	Intronic	A/G	↓	

↑—increased response to therapy (in relation to the minor allele); ↓—reduced response to therapy (in relation to the minor allele; bold font highlights the SNPs, associated with treatment response in 2 and more studies; ^†^—(in relation to the major allele diplotype).

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
