# Peer review of "Pharmacogenetics of Type 2 Diabetes—Progress and Prospects"

_ijms, 2020, doi:10.3390/ijms21186842_

Round 1

Reviewer 1 Report

This is a useful summary of the various genetic variants that influence commonly used therapies in type 2 diabetes. Whilst the pharmacogentics of metformin, meglitinides, DPP-4 inhibitors and GLP-1 receptor agonists are discussed, I am surpsied to see omission of thiazolidinediones (TZD) such as pioglitazone and also the SGLT-2 inhibitors which are now widely used. it is possible that there is not much literature on the pharmacogenetics of the SGLT-2 inhibitors but this should still be included even if just in a paragraph. 

On a stylistic point- the order of discussion of the genetic variants should correspond to the tables. Thus I would suggest the order for discussion after Table 2 should be 4.2 KCNJ11 as already done, 4.3 CYP2C9, 4.4 NOS1AP, 4.5 TCF7L2.

One minor point- 3.1 introduces the protein OCT1. This should be shown as Organic Cationic Transporter 1 (OCT1). Then 3.2 can simply refer to OCT2.

Author Response

Point 1: This is a useful summary of the various genetic variants that influence commonly used therapies in type 2 diabetes.

Response 1: We thank the Reviewer for the positive assessment of our work as well as for the useful comments and suggestions.

Point 2: Whilst the pharmacogentics of metformin, meglitinides, DPP-4 inhibitors and GLP-1 receptor agonists are discussed, I am surpsied to see omission of thiazolidinediones (TZD) such as pioglitazone and also the SGLT-2 inhibitors which are now widely used. it is possible that there is not much literature on the pharmacogenetics of the SGLT-2 inhibitors but this should still be included even if just in a paragraph. 

Response 2: We thank the Reviewer for this valuable comment. We decided not to include thiazolidinediones in the review, because in recent years the use of the TZD has been declining in many countries, including Russia. A section about novel and promising drug SGLT-2 inhibitors was added in the manuscript.

Point 3: On a stylistic point- the order of discussion of the genetic variants should correspond to the tables. Thus I would suggest the order for discussion after Table 2 should be 4.2 KCNJ11 as already done, 4.3 CYP2C9, 4.4 NOS1AP, 4.5 TCF7L2.

Response 3: The order of discussion of the genetic variants was updated according the order of variants in the tables.

Point 4: One minor point- 3.1 introduces the protein OCT1. This should be shown as Organic Cationic Transporter 1 (OCT1). Then 3.2 can simply refer to OCT2.

Response 4: The full name of the protein has been added to the corresponding section of the manuscript.

Reviewer 2 Report

Dear Editor,

The article by Nasykhova A. et al. entitled "Pharmacogenetics of type 2 diabetes: progress and 2 prospects" is a review that analyze the present data and literature on the correlations between different genetic polymorphisms and treatment with metformin, sulfonylureas/meglitinides, DPP-4 inhibitors and GLP-1 receptor agonists. The analysis is very accurate and highlights how the results in different cases can be contradictory. However, some aspects remain to be clarified:

  • Why are SGLT-2 inhibitors not been reported in this review?
  • The discussion should be more extensive andemphasize the potential role of epigenetics and small RNAs such as miRNA in the  modulation of the different reported SNPs.
  • In the tables it could be useful to highlight the polymorphisms (SNPs), which action have been confirmed with more studies and whose data are concordant. Furthermore, it would be important to divide the tables by polymorphisms (SNPs) that increase response to treatment and those that reduce it. All these information would make the tables more readable.

Author Response

Point 1: The article by Nasykhova A. et al. entitled "Pharmacogenetics of type 2 diabetes: progress and 2 prospects" is a review that analyze the present data and literature on the correlations between different genetic polymorphisms and treatment with metformin, sulfonylureas/meglitinides, DPP-4 inhibitors and GLP-1 receptor agonists. The analysis is very accurate and highlights how the results in different cases can be contradictory. However, some aspects remain to be clarified.

Response 1: We thank the Reviewer for the positive assessment of our work as well as for the useful comments and suggestions.

Point 2: Why are SGLT-2 inhibitors not been reported in this review?

We thank the Reviewer for this valuable comment. To date, the information about genetic variants involved in response to SGLT-inhibitors therapy remains quite limited. We added a section about this novel and promising drug and included the latest data in this field.

Point 3: The discussion should be more extensive andemphasize the potential role of epigenetics and small RNAs such as miRNA in the  modulation of the different reported SNPs.

Response 3: The discussion was updated. The additional information including the analysis of different epigenetics mechanisms in treatment response was added.

Point 4: in the tables it could be useful to highlight the polymorphisms (SNPs), which action have been confirmed with more studies and whose data are concordant. Furthermore, it would be important to divide the tables by polymorphisms (SNPs) that increase response to treatment and those that reduce it. All these information would make the tables more readable.

Response 4: We highlighted the SNPs which association has been confirmed in several studies.  We considered it possible to indicate the effect of SNPs in the tables, so that the discussion of SNPs was in accordance with their order in the tables and because some SNPs have been shown by different studies to have the both effects.